# TLR4—A Pertinent Player in Radiation-Induced Heart Disease?

**DOI:** 10.3390/genes14051002

**Published:** 2023-04-28

**Authors:** Basveshwar Gawali, Vijayalakshmi Sridharan, Kimberly J. Krager, Marjan Boerma, Snehalata A. Pawar

**Affiliations:** 1Department of Radiation Oncology, SUNY Upstate Medical University, Syracuse, NY 13210, USA; gawalib@upstate.edu; 2Division of Radiation Health, College of Pharmacy, the University of Arkansas for Medical Sciences, Little Rock, AR 72205, USA; vmohanseenivasan@uams.edu (V.S.); kjkrager@uams.edu (K.J.K.); mboerma@uams.edu (M.B.); 3Upstate Cancer Center, SUNY Upstate Medical University, Syracuse, NY 13210, USA

**Keywords:** cardiac dysfunction, heart, inflammation, ionizing radiation, nitrosative stress, oxidative stress, radiation-induced heart disease, Toll-like receptor 4

## Abstract

The heart is one of the organs that is sensitive to developing delayed adverse effects of ionizing radiation (IR) exposure. Radiation-induced heart disease (RIHD) occurs in cancer patients and cancer survivors, as a side effect of radiation therapy of the chest, with manifestation several years post-radiotherapy. Moreover, the continued threat of nuclear bombs or terrorist attacks puts deployed military service members at risk of exposure to total or partial body irradiation. Individuals who survive acute injury from IR will experience delayed adverse effects that include fibrosis and chronic dysfunction of organ systems such as the heart within months to years after radiation exposure. Toll-like receptor 4 (TLR4) is an innate immune receptor that is implicated in several cardiovascular diseases. Studies in preclinical models have established the role of TLR4 as a driver of inflammation and associated cardiac fibrosis and dysfunction using transgenic models. This review explores the relevance of the TLR4 signaling pathway in radiation-induced inflammation and oxidative stress in acute as well as late effects on the heart tissue and the potential for the development of TLR4 inhibitors as a therapeutic target to treat or alleviate RIHD.

## 1. Introduction

Radiotherapy (RT) is used to treat more than 70% of all cancer types. Improvements in cancer therapy have led to an ever-increasing cancer cure rate. However, both RT and many chemotherapies are associated with side effects due to injury of normal (non-cancer) tissues that are sometimes severe and almost always reduce the quality of life of cancer patients and survivors. From a clinical perspective, the dose of ionizing radiation (IR) that can be delivered is limited by the radiation tolerance of normal tissue [1]. As a result, the impact of these side-effects of RT on the quality of life of cancer survivors due to normal tissue complications is now coming to attention as they represent a growing medical problem [2,3]. An improved understanding of normal tissue injury is therefore required to develop pharmaceutical interventions that increase the safety of cancer therapy [4]. Exposure of the heart to high doses of radiation has been known to cause cardiac dysfunction. In clinic, RT has been widely used for the treatment of thoracic tumors, Hodgkin’s lymphoma, and breast cancer. As every therapy has side effects and limitations, an important side effect is on the cardiac system known as radiation-induced heart disease (RIHD) [5,6,7,8,9,10,11]. Military service members continue to be at risk for exposure to IR because of nuclear warfare, terrorism, or rescue operations after nuclear or radiological events. Moreover, military service members and veterans who are treated for thoracic cancers may also be at risk of cardiac side effects of their cancer treatment.

For decades, the heart was thought to be resistant to radiation doses below those administered in RT [12]. The heart was considered to be safe when less than 10–40% of its volume was exposed to 25 Gy (in conventional RT dose fractions) [12]. However, recent analyses of atomic bomb survivors showed a significant increase in the incidence of cardiovascular disease in individuals exposed to IR (mostly in the form of γ-rays) at doses as low as 2 Gy [13,14,15,16,17]. The estimated radiation-related risk per Gy (ERR Gy^−1^) was 0.14 (95% confidence interval 0.06–0.23) with a linear dose response and doses greater than 0.5 Gy were associated with an elevated risk of stroke and heart disease [16,18,19]. While these studies suggest there is an elevated risk of cardiovascular disease after doses below ~2 Gy, more research in populations of low-dose radiation exposure is required to determine dose-outcome relationships [19]. Additional epidemiological studies on low-dose exposures due to occupation or medical treatment confirm that cardiovascular alterations occur after lower doses of IR than was previously thought, and these findings have re-opened the debate on the radiation sensitivity of the heart [20,21,22,23].

## 2. RIHD

RIHD appears several months to years after RT [6,24,25]. The earliest evidence of RIHD was available from data from cancer patients in the 1980s, the period when a higher range of radiation doses was used to treat cancers and a large part of the heart was exposed to radiation [24,26,27,28,29]. RIHD is characterized by structural and functional pathophysiological abnormal changes to the various parts of the heart like the conduction system, pericardium, myocardium, valves, and coronary vessels [10,30,31]. RIHD manifests different forms of cardiac dysfunction, such as accelerated atherosclerosis, adverse myocardial remodeling, conduction abnormalities, fibrosis, and acute pneumonitis risk [8,24,32,33]. In breast cancer patients receiving a dose of 5 Gy to the left ventricular volume, a 7.4% increase in the rate of coronary-related events was found per Gy mean dose to the heart [34]. When the radiation dose is increased from 5 to 9 Gy and greater than 10 Gy, the risk of developing ischemic heart diseases like acute myocardial injury and angina pectoris doubles [35,36]. 

Acute pericarditis was the first RIHD to appear in thoracic cancer patients who had undergone RT [37]. Patients developed a fever, chest pain, and abnormalities in their electrocardiogram due to an increase in oxidative stress and changes in the conduction system [38]. The early chronic phase is characterized by several compensatory mechanisms such as hypertrophy of cardiomyocytes and proliferation of endothelial cells that manage to survive the initial sub-lethal damage [39]. Insufficiency of these compensatory mechanisms, in turn, drives chronic inflammation and oxidative stress. These processes lead to increased proliferation/activation of fibroblasts which remodel the extracellular matrix causing tissue fibrosis, ultimately in alterations of tissue structure and function [39]. Chronic RIHD includes chronic pericarditis, ischemic heart disease, cardiomyopathy and heart failure, valvular heart disease, and chronic conduction system abnormalities [39,40]. 

Cardiomyopathy is a condition that develops several years after exposure to radiation therapy and is characterized as heart failure with preserved ejection fraction, a characteristic pathological condition that develops in the first phase of heart failure due to compensatory left ventricular hypertrophy and diastolic dysfunction [39,41]. Ten years after RT, there are several structural changes that occur in the heart, such as cardiac fibrosis, and leaflet thickening regurgitation of mitral valves. If untreated, this leads to over 20 years of stenosis and valvular heart disease [30,42]. Chronic conduction system abnormalities are a pathological condition with QTc prolongation, ventricular extrasystole, and atrioventricular block. Changes in the microvessel density, fibrotic lesions, and chronic conduction system abnormalities occur in up to 5% of patients [37]. IR causes a significant increase in the left ventricular wall thickness reducing the decrease in inner wall diameter. This further progresses to diastolic dysfunction, developing heart failure with elevated left ventricular filling pressures and preserved ejection fraction. This stage tends to lead to heart failure with reduced ejection fraction, which in the later stage involves depositions of collagen and cardiac fibrosis [39,41]. 

The molecular mechanisms by which IR injures the heart are largely unknown, and pharmaceutical interventions are currently unavailable. It is crucial to manage the cardiovascular complications of RIHD as it impacts not only the quality of life but also increases the cost of healthcare for cancer survivors and military personnel that may be exposed to IR. An improved understanding of RIHD is therefore necessary for developing pharmaceutical interventions to prevent or treat this complication. 

## 3. Pathophysiology of RIHD

Several studies have demonstrated the role of radiation-induced oxidative/nitrosative stress as the initiating event in RIHD [39]. Exposure to IR results in the radiolysis of water molecules within the cell which causes the generation of reactive oxygen species (ROS), which interact with the cellular lipids, proteins, and deoxyribonucleic acid (DNA) [38,43]. The generation of free radicals results from just a few hours after exposure to radiation; however, its effects can last for several years [44,45]. RT interferes with the normal mitochondrial mechanism thereby disrupting the mitochondrial respiratory chain. Various antioxidant enzymes in the human body, like glutathione reductase, superoxide dismutase (SOD), heme oxygenase, glutathione peroxidase, and catalase, are known to have free radical scavenging abilities, but at this step because of excessive generation of ROS and reactive nitrogen species (RNS), these enzymes are incapable to neutralize excessive ROS [46]. This increase in ROS levels post-IR exposure exacerbates pre-existing imbalances in the antioxidant/oxidant system leading to apoptotic cell death. Similarly, oxidative stress also results in the activation of inflammatory cytokines like Interleukin-1 beta (IL1-β) and tumor necrosis factor-alpha (TNF-α) that interfere with cellular homeostatic processes like the Krebs cycle, cellular Ca^2+^ levels, and metabolism [47]. IR exposure also results in the upregulation of inducible nitric oxide synthase (iNOS) to generate nitric oxide (NO) and promotes radiation-induced tissue injury [48]. Peroxynitrite is produced through the interactions of NO and ROS and constitutes the RNS which causes tissue injury by nitrosylation of tyrosine residues of cellular proteins and rendering them non-functional [49]. The acute phase of RIHD is thus characterized by oxidative and nitrosative damage of the cellular proteins, lipids, and DNA leading to cell death, and results in the release of damage-associated molecular patterns (DAMPs) to stimulate the expression of cytokines and chemokines, which recruit immune cells for clearing the dying/dead cells [39]. 

The vascular system also plays an important role in the pathophysiology of RIHD, which is characterized by senescence and dysfunction of endothelial cells. In vitro studies in human microvascular endothelial cells showed increased levels of pro-inflammatory cytokines intercellular adhesion molecule 1 (ICAM-1), Interleukin-6 (IL-6), and Interleukin-8 (IL-8) in the absence of immune cells after exposure to a single dose of 15 Gy [50]. This shows the relation between the pro-inflammatory state and the vascular endothelium. Activation of pro-inflammatory cytokines and matrix metalloproteinases (MMP), specifically MMP-1 and MMP-2 in endothelial cells by increased oxidative stress, leads to protease enzyme involvement, recruiting macrophages and neutrophils which results in vasoconstriction and tissue hypoxia. Exposure to IR results in endothelial dysfunction, due to inhibition of thrombomodulin, increased levels of von Willebrand factor, resulting in improper fibrinolysis [51]. Various coagulation factors, like thrombin, cause secretion of chemokines like IL-8 and monocyte chemoattractant peptide which results in chemotaxis of neutrophils in a series of events causing vascular inflammation [52]. 

IR causes the induction of endoplasmic reticulum (ER) stress, which involves the unfolded protein response which is caused by an imbalance between correctly or incorrectly folded proteins [53]. ER-stimulated cardiomyocytes cause the release of Ca^2+^ into the cytoplasm which further leads to increased levels of Ca^2+^ in mitochondria and activation of pro-apoptotic proteins like Bax [54], while surviving cardiomyocytes show compensatory hypertrophy [53]. 

Fibrosis is the result of chronic radiation-induced damage and is a dynamic process involving oxidative stress, chronic hypoxia, pro-fibrotic cytokines, and alterations at the phenotypic level [55]. Fibrosis is one of the main processes which cause delayed radiation-induced damage, mainly involving the deposition of collagen by activated fibroblasts. After irradiation, neutrophils secrete pro-fibrotic factors like platelet-derived growth factor (PDGF), transforming growth factor-beta (TGF-β), Interleukin-13 (IL-13), and Interleukin-14 (IL-14). Especially TGF-β has a crucial role in the accumulation of myofibroblasts and inhibition of the enzyme collagenase, which is important for breaking down collagen I and collagen III. Connective tissue growth factor, which is induced by TGF-β causes the differentiation of mesenchymal cells and fibroblasts into pro-fibrotic myofibroblasts [56]. 

Literature review suggests that modification in epigenetics has a significant role in normal tissue injury, specifically fibrosis [57]. Different mechanisms by which epigenetic modifications are achieved are regulation of the non-coding RNA and modification of the histone residue involving acetylation, methylation, phosphorylation, and DNA methylation. Epigenetic reprogramming contributes to radiation-induced fibrosis, particularly via DNA methylation. α-smooth muscle actin (α-SMA), a gene of myofibroblast differentiation, is reported to be regulated by methylation of CpG islands [57]. Fibrosis can be promoted by hypermethylation of genes responsible for apoptosis resulting in decreased cell death [58]. Therefore, methylation-inhibiting agents may be promising new treatment strategies in RIHD. Similarly, irregularity in microRNA activity and histone modifications, which are also types of epigenetic modulation, are linked with fibrosis in the heart and other tissues [57]. 

Interestingly, several clinical and pre-clinical studies have shown a critical role of the innate immune receptor, Toll-like receptor 4 (TLR4), in cardiovascular disease conditions such as acute myocardial infarction, ischemic heart injury, heart failure, dilated cardiomyopathy, and septic shock [59,60,61,62,63,64,65,66]. In this review, we discuss further the role of Toll-like receptors (TLRs) in the heart and specifically the role of TLR4 in the pathophysiology of the heart and its potential role in RIHD.

## 4. Role of TLRs in the Heart 

Discovered in the year 1996 by Lemaitre et al. in the *Drosophila melanogaster*, the TLRs are being explored for their role in various pathological conditions as well as the normal mechanism of the human body [67]. These receptors are linked to innate immunity by playing a key role as pattern recognition receptors (PRR). Activation of PRR is caused by DAMPs or PAMPs, leading to an inflammatory cytokine cascade [68]. TLRs are innate immune pattern recognition receptors expressed in immune cells that respond to extrinsic ligands produced by various pathogens, also termed pathogen-associated molecular patterns (PAMPs), and to intrinsic ligands associated with tissue injury, also known as DAMPs [69]. They are also expressed on endothelial, smooth muscle cells, fibroblasts, and cardiac myocytes and are able to release and sense DAMPs releases in response to cellular stress and tissue injury [70]. Currently, a total of ten TLRs in humans and thirteen TLR families in mice have been identified [71]. Several stimuli that activate the TLRs include endogenous ligands like human cardiac myosin, high-mobility group box 1 protein, DNA, messenger RNA, small interfering RNA, fibronectin, hyaluronan, and heat shock proteins, and exogenous agents, like lipopolysaccharides from Gram-negative bacteria, bacterial flagellin and single-stranded RNA [72,73,74,75,76,77,78,79,80,81]. TLRs are activated by the myeloid differentiation primary response gene 88 (MyD88) and Toll-IL1R domain-containing adapter (TIR) and induce transcription of genes required for inflammatory responses. While signaling via TLRs plays a key role in innate immune cells, such as macrophages, to induce inflammation for pathogen clearance, they can also propagate inflammation in response to tissue injury and play a key role in cardiovascular diseases [59]. 

TLR2 is highly expressed in cardiac myocytes, and its over-activation is implicated in myocardial fibrosis, cardiac dysfunction, activation of apoptosis, and cardiac inflammation [82]. Studies demonstrate the involvement of ROS in TLR2 activation [83]. TLR3 is involved in the regeneration of cardiomyocytes in neonates and plays a crucial role in the activation of NF-kB and Interferon-regulatory factors 3 and 7 [84]. Studies in TLR3-knockout mice revealed that reduced expression of caspase 2 and 3 led to improved cardiac function [84]. TLR3 is an endogenous sensor of necrosis during acute inflammation and an inducer of necroptosis via the TIR-domain-containing adapter-inducing interferon-β (TRIF)-RIP1 pathway [73,85]. TLR3 plays a deleterious role in cardiac dysfunction during polymicrobial sepsis and in animal models of myocardial infarction [86,87]. TLR5 promotes damage by increased expression of inflammatory mediators, such as IL-6, monocyte chemoattractant protein 1, and macrophage inflammatory protein 2 in cardiac dysfunction. TLR7 takes part in nucleotide sensing of the Toll/IL-1 family and is involved in the cardiac inflammation process via activation of MyD88 and TNF receptor-associated factor 6 (TRAF6), resulting in nuclear factor-κB (NF-κB) mediated inflammatory response [88,89]. TLR4 is expressed in different innate immune cells and mainly in heart cardiomyocytes, vascular smooth muscle cells, and fibroblasts [90]. TLR4 causes the activation of inflammatory pathways via the activation of leukocytes, and secretion of cytokines, leading to apoptosis and necrosis of cardiomyocytes [91,92]. Among TLR4 adaptor molecules, MyD88 was the first adaptor molecule discovered to be critical for TLR signaling (MyD88-dependent pathways) [93]. Activated TLR4 signals through the canonical NF-κB pathway increase the expression of pro-inflammatory cytokines, such as TNF-α and interleukin-1β (IL-1β) [94]. 

## 5. TLR4 Upregulation Is Associated with Heart Pathologies 

Inflammation plays a critical role in the case of several types of cardiac dysfunction [95]. Dysregulation of TLR4 signaling plays a critical role in the initiation and/or progression of cardiovascular diseases, such as ischemia-reperfusion injury, atherosclerosis, myocardial infarction, heart failure, dilated cardiomyopathy, hypertension, sepsis neuropsychiatric, and neurodegenerative disorders [64,66,96,97,98,99,100,101,102,103,104]. For instance, the upregulation of TLR4 causes inflammation and aggravates heart failure [105,106]. Another pathophysiological condition associated with increased TLR4 activation is cardiac arrhythmias [64]. It was shown that activation of TLR4 decreased the action potential duration of cardiomyocytes via the IRF-3 pathway, a MyD88-independent pathway that leads to cardiac arrhythmias [107]. The expression levels of TLR4 mRNA and protein were found to be elevated in mononuclear cells of patients with coronary artery disease, acute myocardial ischemia, atherosclerosis, acute myocardial infarction, heart failure, and dilated cardiomyopathy [62,108,109,110]. TLR4 upregulation was also associated with high levels of ROS and increased production of iNOS that led to damage of the myocardium and to myocardial ischemia-related injuries [111]. In another study, it was found that patients with a history of acute myocardial ischemia showed increased plasma levels of TLR4, which correlated with elevated expression of inflammatory cytokines, such as IL-6, IL-2, IL-8, IL-10, and TNF-α [102,112]. Reports also suggest the involvement of TLR4 in the pathophysiology of acute cardiac dysfunction, which is caused by septic shock as well as myocardial ischemia. Chimeric mice treated with TLR4-deficient leukocytes show a marked reduction in the function of myocardial contraction when treated with lipopolysaccharide (LPS). This demonstrates the involvement of TLR4 in the leukocytes in promoting myocyte dysfunction [103]. Recent studies in patients with SARS-CoV-2 infection also revealed activation of TLR4 in the heart and lungs to cause aberrant TLR4 signaling via the MyD88-dependent pathway that contributes to acute lung injury, myocarditis, cardiac complications, and other severe inflammatory complications, such as the hyper inflammation observed in patients with severe COVID-19 [113].

## 6. TLR4 Overexpression or Knockdown in Preclinical Models of Cardiovascular Diseases

TLR4 is expressed and elevated following ischemic injury in both leukocytes and cardiac myocytes [99]. Studies in isolated hearts and cardiomyocytes show increased apoptosis mediated by TLR4 in an ischemia-reperfusion model [82]. TLR4-mediated immune cell infiltration and inflammatory cytokines are implicated in the hypertrophy of cardiomyocytes [114]. Mice treated with a TLR4 inhibitor showed a significant decrease in all these parameters, which shows the involvement of TLR4 receptor signaling in cardiac hypertrophy [115]. LPS treatment of rat cardiomyocytes H9C2 in vitro and male Sprague Dawley (SD) rats in vivo revealed increased expression of TLR4 expression, caspase-7/caspase-9, oxidative stress, fibrosis markers, mitochondrial complex activity, and decreased levels of SIRT-2 protein [116]. These findings demonstrate that activation of TLR4 by LPS results in apoptosis, cardiac inflammation, mitochondrial dysfunction, and fibrosis both in vitro and in vivo [116]. TLR4 upregulation in cardiomyocytes is also implicated in a mouse model of doxorubicin-induced dilated cardiomyopathy [117]. An isoproterenol-induced cardiac hypertrophy model of male SD rats showed a significant increase in TLR4 expression along with mitochondrial dysfunction. A recent experimental study has reported that monocyte TLR4 expression is upregulated in patients with heart failure after acute myocardial infarction. The LPS treatment of chimeric mice that received TLR4−/− immune cells, showed significant reduction in the ventricular myocytes shortening. These results suggest that immune cells TLR4 promote cardiac dysfunction in an endotoxemia model [103]. Loss of TLR4 in mice attenuated-aging-associated cardiovascular dysfunction by suppressing oxidative stress by upregulation of Cu-Zn superoxide dismutase (SOD1) and Manganes-SOD (SOD2) and downregulation of senescence markers p16 and p53 [118]. 

TLR4 inhibitors or knockdown approaches have been shown to reduce cardiac injury in many animal models. Preclinical studies with TLR4 inhibition or gene knock-out approaches resulted in reduced cardiac injury in animal models, including cardiac dysfunction due to trauma-hemorrhage models [59,119]. In a model of hypertension, male SD rats treated with a TLR4 inhibitor showed a significant reduction in myocardial inflammation and expression of iNOS. These results show that TLR4 blockade inhibits the hypertensive response causing downregulation in the myocardial inflammatory activity [120]. Knockdown of TLR4 showed significant improvement in left ventricular remodeling and reduced cell death of cardiomyocytes [106]. Another study revealed reduced infarct size in the case of TLR4-deficient (C3H/HeJ) mice [96]. In myocardial injury models, it was found that TLR4-deficient mice had smaller infarctions and decreased damage to the myocardium, indicating that TLR4 plays a key role in promoting myocardial damage [121]. In the case of TLR4-deficient mice, all the autophagy genes Beclin-1, Atg7, Atg-5, LC-2, and α-SMA were found to be downregulated [106,122]. In spontaneously hypertensive rats, it was shown that silencing of TLR4 inhibited atrial fibrosis and susceptibility to atrial fibrillation through downregulation of the Nod-like receptors’ family pyrin domain containing 3 (NLRP3)-TGF-β [123]. 

## 7. Role of TLR4 in RIHD in Rodent Models

Different types of radiation approaches are being applied in pre-clinical settings like partial heart irradiation, whole thorax, and whole heart radiation with radiation delivery methods like conformal systems and single beam with lead shielding [124]. Recent studies in rodent models have confirmed that adverse cardiac remodeling as delayed effects of acute radiation exposure (DEARE) after total body irradiation (TBI) or partial body irradiation (PBI) can indeed be reproduced in animal models [125,126,127,128,129]. Inflammation is thought to play an important role in the development of RIHD [7]. However, very few studies have examined the effect of IR on the expression of TLR4 in the heart. Some recent studies in mouse models revealed a key role of TLR4 activation in acute as well as delayed adverse effects of IR-promoting pulmonary fibrosis, radiation pneumonitis, radiation-induced liver disease, and radiation-induced oral mucositis [130,131,132,133,134]. Another study using a combination of immunotherapy using the PD-1 inhibitor and RT revealed upregulation of TLR4 in mouse hearts [135]. Studies from our group revealed upregulation of left ventricular expression of TLR4 in both male and female C57BL/6 mice at 6 months after exposure to 9.5 Gy gamma rays in a mouse model where both the hind limbs were shielded from radiation to promote long-term survival and allow the study of DEARE [136]. We recently reported an elevated expression of TLR4 in mouse hearts at 6 months post-local heart or partial heart irradiation which correlated with increased markers of cardiac fibrosis [137]. 

Since previously published studies by us and others consistently show an upregulation of TLR4 in the irradiated heart, we hypothesize that TLR4 upregulation may play an essential role in IR-induced inflammation and oxidative stress in promoting RIHD (Figure 1). Since the heart tissue comprises a significant number of fibroblasts, cardiomyocytes, endothelial, and immune cells, it will be critical to identify the specific cell types involved in dysregulated TLR4 signaling and their role in the development of the sequelae of RIHD. Apart from this, further investigations are also needed to identify the specific type of DAMPs generated in response to radiation that promote the upregulation of TLR4. Radiation induces DAMPs, such as high mobility group box 1 (HMGB1), calreticulin, adenosine triphosphate, and heat shock proteins (HSPs) [138,139]. Apart from the well-known TLR4 ligand-lipopolysaccharide (LPS), DAMPs, such as HMGB1, heat shock factors 60, HSP70, fibrinogen, fibronectin, hyaluronan that are also implicated in cardiovascular diseases, are reported to play a role in the activation of TLR4 [77,82,140,141].

In addition, as it is well-known that inflammation in the early time points post-IR may be essential to recruit immune cells for clearing dying/damaged cells, further investigations will be needed to address the optimal timing for administration of TLR4 inhibitors or antagonists post-radiation exposure in order to see beneficial effects on alleviating RIHD. Interestingly, studies with TLR2 agonists (bacterial lipoprotein, CBL502) and TLR4 agonists (LPS, or its derivatives) treated prior to radiation showed significant protection against acute injury to hematopoietic and intestine tissues [141,142,143,144,145,146,147]. However, the role of these agonists in the context of DEARE has not been investigated and whether they may impart radioprotective effects needs further investigation. In the case of Doxorubicin-associated cardiotoxicity in human patients, it was also observed that elevated serum levels of TLR4 correlated with increased myocardial iron load and left ventricular dysfunction with decreased ejection fraction [148]. In the case of RIHD, a similar follow-up of patients prior to and post-RT is needed to determine whether elevated serum levels of TLR4 correlate positively with delayed adverse effects on the heart. 

## 8. Conclusions 

RIHD remains an important complication of cancer RT regimens. The only available approach currently is to limit complications from RT through efforts to reduce cardiac exposure to IR. Despite technological improvements in treatment planning and delivery of RT, patients with cancers in the lung, esophagus, and proximal stomach can still receive significant doses of IR to the heart and continue to be at risk of IR-induced cardiomyopathy [149,150,151]. Currently, there are no approved radiation countermeasures to alleviate or prevent RIHD. 

Various pre-clinical and clinical models have shown the importance of TLR4 in promoting inflammation and oxidative stress in cardiovascular diseases. Similar work by us and others show that mouse models of localized radiation therapy as well as TBI resulted in elevated expression of TLR4 in mouse hearts and correlated with markers of adverse tissue remodeling and cardiac dysfunction. This suggests that TLR4 antagonists or inhibitors may be suitable to block the downstream effects of the TLR4/MyD88-dependent pathway in inflammation and oxidative/nitrosative stress and alleviate the radiation-related adverse effects on the heart (Figure 1). Further research is needed to identify the mechanistic role of the TLR4 pathway and the optimal timing for the administration of a TLR4 antagonist or inhibitor to treat or prevent RIHD. The effects of TLR4 inhibition on other organ systems also need to be assessed before the clinical development of TLR4 inhibitors for treatments of RIHD.


## Figures and Tables

**Figure 1 genes-14-01002-f001:**
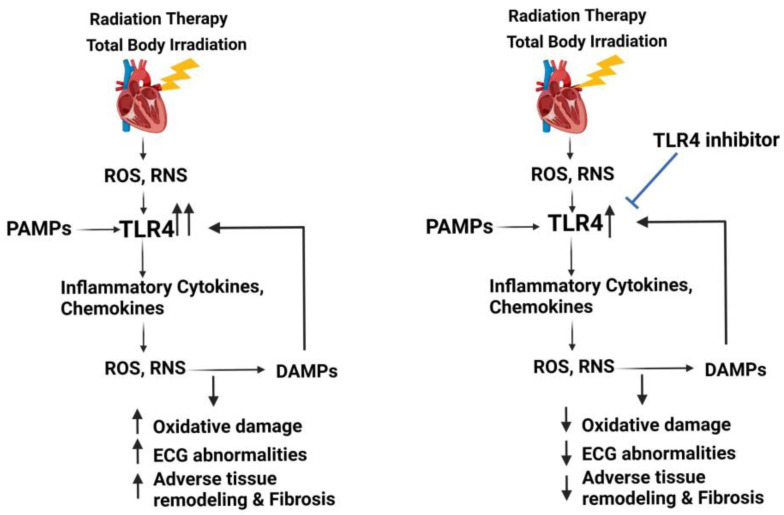
Potential role for TLR4 as a therapeutic target to alleviate RIHD. Left: hypothesized role of the TLR4 pathway in the development of RIHD. Right: TLR4 inhibition may interfere with the events of adverse tissue remodeling in RIHD.

## Data Availability

Not applicable.

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
