# Peer review of "TLR4—A Pertinent Player in Radiation-Induced Heart Disease?"

_genes, 2023, doi:10.3390/genes14051002_

Round 1

Reviewer 1 Report

Gawali et al present a very well-written review of radiation-induced heart disease and hypothesize that observations of increased Toll-like receptor expression in heart tissue and plasma from irradiated animals contribute to RIHD.

The review is concise and summarizes current literature well. That said, there are a couple of major omissions that should be addressed:

1) Clear definitions of doses and exposures and those used in the cited literature are lacking.

Specifically, the use of the term “low” and “high” dose needs to be defined and related to the literature reviewed. For example p3L90 states “Exposure to IR results in…”. What are the dose thresholds needed to observe these phenomena robustly. The authors highlight increased risk of non-cancer cardiovascular disease incidence in atomic bomb survivors and nuclear industry workers as if it were the consensus in the field. It is not. It is a large stretch to relate these contexts to RIHD, which is better understand and accepted as a consequence of clinical radiation interventions rather than a result of environmental exposure.

p2. In the first paragraph there is no reference to literature claiming a dose-dependent effect on the “severity of heart disease”. How would severity be measured? Do they mean incidence? Not clear. Same with doubling of risk (ERR?). 

2) There is some literature regarding the radioprotective effect of TLR4 expression (and possibly other TLRs?). Given that the major proportion of RIHD morbidity/mortality is due to radiotherapies, it would seem important and appropriate to broach this topic in this review. Might TLR induction prior to RT be protective for bystander/abscopic cells?

3) given the above (I.e., current body of knowledge) Figure 1 seems a bit too simplistic. Would TLR4 ablation remove protective effects that might be expressed across a RT regimen? Should the removal of TLR4 on the right side of the panel be tissue specific?

Minor issues/typos:

The authors make reference to unpublished data from their laboratory. I’m not sure what the journal policy is regarding this but this is generally discouraged in a literature review. If the work will be published imminently (minimally as a preprint) this should be done prior to acceptance of this work.

L83: remove “and”

L100: “Krebs cycle”

L126: what doses?

L164: “plays a key role”

L171: neonatal level? Do they mean “in neonates”. Please elaborate 

Author Response

On behalf of all the authors, I thank  the reviewer for their very favorable and insightful comments which have helped to improve the review article under consideration.  In the revised manuscript, we have addressed all the points raised by the reviewers to the best extent possible. The description of changes are listed below in our response to the Reviewer’s comments.

Reviewer 1 report:

Gawali et al. present a very well-written review of radiation-induced heart disease and hypothesize that observations of increased Toll­ like receptor expression in heart tissue and plasma from irradiated animals contribute to RIHD.

The review is concise and summarizes current literature well. That said, there are a couple of major omissions that should be addressed:

  • Clear definitions of doses and exposures and those used in the cited literature are lacking.

  • Specifically, the use of the term "low" and "high" dose needs to be defined and related to the literature reviewed.

Response:   We thank the reviewer for pointing this important point. We have included definitions of what we consider low and high doses. Since there is some controversy about the development of RIHD at doses below ~2 Gy, we refer to high radiation doses when they are above 2 Gy (Simon et al., 2022, Little et al., 2021). We have included this sentence in the revised manuscript (Page 2, L44-L51).

  • For example p3L90 states "Exposure to IR results in... ". What are the dose thresholds needed to observe these phenomena robustly.

Response:  For the phenomenon of local IR-induced reactive oxygen species formation, to our knowledge there is no dose threshold. (Page 3 L101 in the revised manuscript)

  • The authors highlight increased risk of non-cancer cardiovascular disease incidence in atomic bomb survivors and nuclear industry workers as if it were the consensus in the field. It is not. It is a large stretch to relate these contexts to RIHD, which is better understand and accepted as a consequence of clinical radiation interventions rather than a result of environmental exposure.

Response:   We agree with the reviewers’comment.  We have included the doses to define “low and or high doses of radiation” and provided more detailed information on findings in the A-bomb survivors. We have added the sentence “While these studies suggest there is an elevated risk of cardiovascular disease after doses below ~2 Gy, more research in populations of low-dose radiation exposure is required to determine dose-outcome relationships (Simon et al., 2022, Little et al., 2021) (revised manuscript: Page 2, L49-L51)

  1. Simon, S.L.; Kendall, G.M.; Bouffler, S.D.; Little, M.P. The Evidence for Excess Risk of Cancer and Non-Cancer Disease at Low Doses and Dose Rates. Radiat Res 2022, 198, 615-624, doi:10.1667/rade-22-00132.1.
  2. Little, M.P.; Azizova, T.V.; Hamada, N. Low- and moderate-dose non-cancer effects of ionizing radiation in directly exposed individuals, especially circulatory and ocular diseases: a review of the epidemiology. Int J Radiat Biol 2021, 97, 782-803, doi:10.1080/09553002.2021.1876955.

  • In the first paragraph there is no reference to literature claiming a dose-dependent effect on the "severity of heart disease". How would severity be measured? Do they mean incidence? Not clear. Same with doubling of risk (ERR?).

Response:   We thank the reviewer for pointing this critical point. We have removed the statement on severity of heart disease and instead described the manifestations of RIHD more carefully (revised manuscript: Page 2 L54-65). We have clarified the dose-dependent effects on the incidence of heart disease and ERR in the A-bomb survivors and cited relevant references (Ozasa et al., 2012; Ozasa et al. 2016) (revised manuscript: Page 2 L46-L49).

  1. Ozasa, K.; Takahashi, I.; Grant, E.J. Radiation-related risks of non-cancer outcomes in the atomic bomb survivors. Annals of the ICRP 2016, 45, 253-261, doi:10.1177/0146645316629318.
  2. Ozasa, K.; Shimizu, Y.; Suyama, A.; Kasagi, F.; Soda, M.; Grant, E.J.; Sakata, R.; Sugiyama, H.; Kodama, K. Studies of the mortality of atomic bomb survivors, Report 14, 1950-2003: an overview of cancer and noncancer diseases. Radiat Res 2012, 177, 229-243.

2) There is some literature regarding the radioprotective effect of TLR4 expression (and possibly other TLRs). Given that the major proportion of RIHD morbidity/mortality is due to radiotherapies, it would seem important and appropriate to broach this topic in this review. Might TLR induction prior to RT be protective for bystander/abscopic cells?

Response:  We thank the reviewer for this suggestion. In the revised manuscript, we have now included and discussed the literature that demonstrates radioprotective roles of TLR4 and TLR2 agonists (revised manuscript: Page 8, L320-L324). It is plausible that TLR induction may be protective for the bystander/abscopic cells.

  1. Neta, R.; Oppenheim, J.J.; Schreiber, R.D.; Chizzonite, R.; Ledney, G.D.; MacVittie, T.J. Role of cytokines (interleukin 1, tumor necrosis factor, and transforming growth factor beta) in natural and lipopolysaccharide-enhanced radioresistance. J Exp Med 1991, 173, 1177-1182.
  2. Krivokrysenko, V.I.; Toshkov, I.A.; Gleiberman, A.S.; Krasnov, P.; Shyshynova, I.; Bespalov, I.; Maitra, R.K.; Narizhneva, N.V.; Singh, V.K.; Whitnall, M.H.; et al. The Toll-Like Receptor 5 Agonist Entolimod Mitigates Lethal Acute Radiation Syndrome in Non-Human Primates. PloS one 2015, 10, e0135388, doi:10.1371/journal.pone.0135388.
  3. Shakhov, A.N.; Singh, V.K.; Bone, F.; Cheney, A.; Kononov, Y.; Krasnov, P.; Bratanova-Toshkova, T.K.; Shakhova, V.V.; Young, J.; Weil, M.M.; et al. Prevention and Mitigation of Acute Radiation Syndrome in Mice by Synthetic Lipopeptide Agonists of Toll-Like Receptor 2 (TLR2). PloS one 2012, 7, e33044, doi:doi:10.1371/journal.pone.0033044.
  4. Singh, V.K.; Ducey, E.J.; Fatanmi, O.O.; Singh, P.K.; Brown, D.S.; Purmal, A.; Shakhova, V.V.; Gudkov, A.V.; Feinstein, E.; Shakhov, A. CBLB613: a TLR 2/6 agonist, natural lipopeptide of Mycoplasma arginini , as a novel radiation countermeasure. Radiat Res 2012, 177, 628-642.
  5. Liu, Z.; Lei, X.; Li, X.; Cai, J.M.; Gao, F.; Yang, Y.Y. Toll-like receptors and radiation protection. Eur Rev Med Pharmacol Sci 2018, 22, 31-39, doi:10.26355/eurrev_201801_14097.
  6. Feng, Z.; Xu, Q.; He, X.; Wang, Y.; Fang, L.; Zhao, J.; Cheng, Y.; Liu, C.; Du, J.; Cai, J. FG-4592 protects the intestine from irradiation-induced injury by targeting the TLR4 signaling pathway. Stem Cell Res Ther 2022, 13, 271, doi:10.1186/s13287-022-02945-6.
  7. Xu, Y.; Chen, Y.; Liu, H.; Lei, X.; Guo, J.; Cao, K.; Liu, C.; Li, B.; Cai, J.; Ju, J.; et al. Heat-killed salmonella typhimurium (HKST) protects mice against radiation in TLR4-dependent manner. Oncotarget 2017, 8, 67082-67093, doi:10.18632/oncotarget.17859.

3) given the above (i.e., current body of knowledge) Figure 1 seems a bit too simplistic. Would TLR4 ablation remove protective effects that might be expressed across a RT regimen? Should the removal of TLR4 on the right side of the panel be tissue specific?

Response:  In Figure 1, with the X through TLR4 we tried indicating TLR4 inhibition, not TLR4 removal. This was unclear. The figure has been revised to depict our thinking of using a TLR4 inhibitor to modulate the TLR4 pathway and dampen downstream events of inflammation, oxidative/nitrosative stress and generation of DAMPs that mediate RIHD (see Page 9: revised Figure 1).

We have also added a sentence in the Conclusion section: “The effects of TLR4 inhibition on other organ systems also need to be assessed before clinical development of TLR4 inhibitors for treatments of RIHD."(revised manuscript: Page 9, L338-339).

Minor issues/typos:

  • The authors make reference to unpublished data from their laboratory. I'm not sure what the journal policy is regarding this but this is generally discouraged in a literature review. If the work will be published imminently (minimally as a preprint) this should be done prior to acceptance of this work.

Response:   We have removed the statements on our unpublished studies, as they are still in very early stages for a preprint.

  • L83: remove "and" L100: "Krebs cycle" L164: "plays a key role"

Response: We have made the corrections in the revised manuscript.

  • L126: what doses?

Response: We have modified the statement in the revised manuscript (revised manuscript: Page 4, L139-L144).

  • L171: neonatal level? Do they mean "in neonates". Please elaborate

Response:  We meant to say that TLR3 plays a critical role in the regeneration of cardiomyocytes in neonates.  We have modified the statement (revised manuscript: Page 5, L187-L189).

Reviewer 2 Report

This review article is very well written, but one major question remains unsolved. Agonists of TLR3, TLR7, and TLR9 in endosomes are mainly DAMPs, whereas agonists of TLRs at the plasma membrane are mainly PAMPs; TLR4 is also at the plasma membrane and its agonist, LPS, is well known. What is not clear in this review is what TLR4 agonist is in RIHD? Even if it is not well identified, it is necessary to discuss the issue of the unidentified agonist and to speculate about its identity based on the results of previous studies.

As for TLR3, it is worth mentioning that it is both a sensor of necrosis and an inducer of necroptosis via the TRIF-RIP1 pathway.

Also, please proofread the spelling of "endogenous" in line 172.

Additionally, please spell out "delayed effects of acute radiation exposure" for DEARE.

Author Response

On behalf of all the authors, I  thank both the reviewer for their very favorable and insightful comments which have helped to  improve the review article under consideration.  In the revised manuscript, we have addressed all the points raised by the reviewers to the best extent possible. The description of changes are listed below in our response to the Reviewer’s comments.

This review article is very well written, but one major question remains unsolved.

  • Agonists of TLR3, TLR7, and TLR9 in endosomes are mainly DAMPs, whereas agonists of TLRs at the plasma membrane are mainly PAMPs; TLR4 is also at the plasma membrane and its agonist, LPS, is well known. What is not clear in this review is what TLR4 agonist is in RIHD? Even if it is not well identified, it is necessary to discuss the issue of the unidentified agonist and to speculate about its identity based on the results of previous studies.

Response:  We agree that this is an important point to clarify the role of DAMPs in TLR4 activation. While LPS is the primary agonist of TLR4, several studies have reported that DAMPs such as HMGB1, HSP60, HSP72, fibrinogen and fibronectin that are implicated in cardiovascular diseases, can also bind to TLR4 and promote its activation.  While the exact DAMP implicated in RIHD is not known, the above DAMPs could be involved, but this needs further investigation. We have briefly discussed the role of IR-induced DAMPs that may promote activation of TLR4 in RIHD and cited relevant references  (revised manuscript: Page 8, L307-314). We have included the following paragraph:

“Apart from this, further investigations are also needed to identify the specific type of DAMPs generated in response to radiation that promote the upregulation of TLR4.  Radiation induces DAMPs such as high mobility group box 1 (HMGB1), calreticulin (CRT), adenosine triphosphate (ATP), and heat shock proteins (HSPs) [1,2]. Apart from the well known TLR4 ligand-lipopolysaccharide (LPS), DAMPs such as HMGB1, heat shock factors 60, HSP70, fibrinogen, fibronectin, hyaluronan that are implicated in cardiovascular diseases are reported to play a role in activation of TLR4 [3-5].”

  1. Schaue, D.M., E. D.; Ratikan, J. A.; Xie, M. W.;Cheng, G.;McBride, W. H. Radiation and inflammation. Semin Radiat Oncol 2015, 25, 4-10, doi:10.1016/j.semradonc.2014.07.007.
  2. Ratikan, J.A.; Micewicz, E.D.; Xie, M.W.; Schaue, D. Radiation takes its Toll. Cancer letters 2015, 368, 238-245, doi:10.1016/j.canlet.2015.03.031.
  3. Klee, N.S.; McCarthy, C.G.; Martinez-Quinones, P.; Webb, R.C. Out of the frying pan and into the fire: damage-associated molecular patterns and cardiovascular toxicity following cancer therapy. Therapeutic Advances in Cardiovascular Disease 2017, 11, 297-317, doi:10.1177/1753944717729141.
  4. Bolourani, S.; Brenner, M.; Wang, P. The interplay of DAMPs, TLR4, and proinflammatory cytokines in pulmonary fibrosis. J Mol Med (Berl) 2021, 99, 1373-1384, doi:10.1007/s00109-021-02113-y.
  5. Lin, L.; Knowlton, A.A. Innate immunity and cardiomyocytes in ischemic heart disease. Life sciences 2014, 100, 1-8.

  • As for TLR3, it is worth mentioning that it is both a sensor of necrosis and an inducer of necroptosis via the TRIF-RIP1 pathway.

Response:  We thank the reviewer for this comment and included the suggested statement in the revised manuscript and cited the relevant reference (Page 5, L191-L192).

Kaiser, W.J.; Sridharan, H.; Huang, C.; Mandal, P.; Upton, J.W.; Gough, P.J.; Sehon, C.A.; Marquis, R.W.; Bertin, J.; Mocarski, E.S. Toll-like receptor 3-mediated necrosis via TRIF, RIP3, and MLKL. J Biol Chem 2013, 288, 31268-31279, doi:10.1074/jbc.M113.462341.

  • Also, please proofread the spelling of "endogenous" in line 172.

Response:  The spelling has been corrected (revised manuscript: Page 5, L191).

  • Additionally, please spell out "delayed effects of acute radiation exposure" for DEARE.

Response:  We have spelled out the abbreviation DEARE (revised manuscript: Page 7, L281).